Application of Self-Organizing Maps to characterize subglacial bedrock properties based on gravity, magnetic and radar data – An example for the Wilkes and Aurora Subglacial Basin region, East Antarctica

Jonas Liebsch<sup>1, \*</sup>, Jörg Ebbing<sup>1</sup>, Kenichi Matsuoka<sup>2</sup>

Correspondence to: Jörg Ebbing (joerg.ebbing@ifg.uni-kiel.de)

Abstract. Subglacial bedrock properties are key to understanding and predicting the dynamics and future evolution of the Antarctic Ice Sheet. However, the ice sheet bed is largely inaccessible for direct sampling, and characterization of subglacial properties has so far relied on expert interpretation of airborne geophysical data. To reduce subjective choices in the joint analysis of data and related biases, we present a Self-Organizing Map (SOM), an unsupervised machine learning technique. The concept of SOMs is briefly introduced and we discuss data selection and their associated attributes. First, we analyzed the correlation between attributes to provide a validation of an appropriate choice. Next, we trained the SOM on attributes derived from gravity, magnetics and ice-penetrating radar data for the Wilkes and Aurora Subglacial Basin region in East Antarctica. In contrast to earlier studies, our approach uses original line data as much as possible. These have a much higher resolution than the smooth gridded products used in previous studies. Our results show marked agreement with past studies on predicting regional bed characteristics such as the presence of crystalline basement and sedimentary basins. Additionally, our results indicate the ability to resolve finer details, demonstrating the potential in applying SOM to subglacial geologic mapping.

<sup>&</sup>lt;sup>1</sup> Institute of Geosciences, Kiel University, Germany

<sup>&</sup>lt;sup>2</sup> Norwegian Polar Institute, Tromsø, Norway

<sup>\*</sup> Now at: University of Iceland, Reykjavik, Iceland

# 1 Introduction

Subglacial bedrock properties, are one of the key components in an improved understanding of the Antarctic Ice Sheet (e.g., Aitken et al. 2023, Bingham et al. 2012, Bell et al. 2008, Jordan et al. 2023, McCormack et al. 2022). The properties at the icerock interface can have a significant impact on ice flow dynamics, as they control bed roughness and consolidation, hydrogeological processes, friction, and basal sliding, all of which influence ice flow velocities (Koellner et al., 2019). Of particular interest is sediment layer and sedimentary rocks at the ice base, as these can affect basal friction, water flow and geothermal heat advection (e.g., Koellner, et al. 2019, Zoet and Iverson, 2020, Li et al. 2022, Aitken et al. 2023).

There are very few reflection seismic lines on the Antarctic continent suitable for resolving the upper crust (e.g. Anandakrishnan et al. 1998, Bayer et al. 2009, Leitchenkov et al. 2016). Therefore, geological models are conventionally based on interpretation of bed topography (e.g., Taylor 1914, Elliot, 1975, Jordan et al. 2020), aeromagnetic or airborne gravity datasets (e.g., Ferraccioli et al. 2002, 2009, 2011, McLean et al. 2009, Aitken et al. 2014, Forsberg et al. 2019), or a combination of those (e.g., Li et al., 2023, Wu et al. 2023). Aeromagnetic data are especially well-suited for inferring subglacial geology (Betts et al. 2024). The interpretation of potential field data requires constraints and the combination with other geophysical or petrophysical data sets in an integrated manner is a common choice (e.g., Jordan et al. 2023, Lowe et al. 2024a, b).

Airborne radar data are complementary and well-suited for imaging within the ice but are almost entirely reflected at the icerock interface. Therefore, radar can provide information on the bed-ice interface, while the physical properties of the bedrock itself are difficult to derive. Still, detailed morphology and inferred attributes like roughness can be indicative of some near-surface geological characteristics (e.g., Shepherd et al. 2006, Rippin et al., 2014; Jordan et al., 2010, Jordan et al. 2023). For example, an area with elevated roughness can be inferred to have a more erosion-resistant bed. To complicate the matter, the current and past flow speed of the ice sheets also impacts erosion and modifies the roughness (Jamieson et al., 2014). Hence, a combination of these data sets might overcome some of the limitations.

Recently, Aitken et al. (2023) presented a detailed classification of geological bed type in Antarctica by analyzing multiple geophysical data sets and models. Hereby, they compiled and synthesized available data and models into a classification map. While Aiken et al. (2023) presented continent-wide, detailed classification of geological bed types, it is an interpretation and remains equivocal at some locations due to complex geology and/or limited data coverage. Another limitation is that the compilation is based on earlier interpretations, which have used different methods and are partly relying on expert knowledge (Aitken et al. 2023).

Machine learning and statistical methods are becoming popular approaches to reduce bias in models. Examples are estimates of geothermal heat flow (Lösing and Ebbing, 2021, Stål et al. 2021), the presence of sedimentary rocks (Li et al., 2022) or subglacial geology (MacGregor et al. 2024). Statistical methods like Stål et al. (2021) and MacGregor et al. (2024) identify boundaries from multiple geophysical and geological datasets. Heterogeneous data coverage and different resolutions of the underlying models limit the validity, especially on a survey scale. As an alternative, machine learning methods such as gradient boosting regression tree have especially become popular to map subglacial properties in both Greenland and Antarctica (e.g., Rezvanbehbahani et al. 2019, Lösing & Ebbing, 2021, Li et al. 2022, Colgan et al. 2023). As for the statistical methods, these

studies are commonly on the scale of an entire continent and rely on training datasets of reasonable size, and the subjective choice of input data.

As another alternative, we employ here Self-Organizing Maps (SOMs; e.g., Kohonen, 1990, Klose 2006) to exploit local, geophysical information on a survey scale. SOMs are an unsupervised machine learning approach that estimates similarities within different data types without assigning them to predefined categories. In order to detect the similarities, a number of attributes is provided based on the initial datasets. These are ideally chosen to enhance the feature of interest, here details in subglacial geology. The advantage is that no training dataset is needed, the produced map can be used for geological interpretation, and the importance of the input attributes can be assessed. To present the possibilities of using SOMs, we chose parts of the Wilkes and Aurora Subglacial Basins in East Antarctica (Figure 1). The area is a key region for studying the role of tectonic boundary conditions on the behavior of the East Antarctic Ice Sheet (Aitken et al. 2014, McCormack et al. 2022), but is chosen here mainly for the quality and preprocessing of the survey data. In the following, we will shortly summarize the concept of SOMs and introduce the data and attributes used for our analysis. We discuss our results both in comparison to the classification by Aitken et al. (2023), and with respect to the choice of input data.

Figure 1: Overview of study area in East Antarctica: A) Bed elevation from Bedmachine (Morlighem, 2020, 2022) with main subglacial and geographical features annotated. B) Geological bed types from the study by Aitken et al. (2023). The black lines indicate the flight paths of the surveys used in our study.

# 2 Self-Organizing Maps

SOMs, unlike other unsupervised learning algorithms, do not attempt to categorize data; rather, they reduce the dimensionality of complex datasets. In our example, we will map the three datasets (bed elevation, gravity, and magnetics) and the related attributes into a 2D space (map) representation. In this space, similar data points are placed in proximity to each other, enabling the identification of clusters. In the following, we briefly explain the concept of SOMs. More details and examples for geological mapping can be found, for example, in Klose (2006) or Carneiro et al. (2012). Self-Organizing Maps (SOMs; e.g., Kohonen 1990) are a simple neural network consisting of a single layer. Each neuron represents a cell on the two-dimensional map with one weight for every dimension of the input data. Neuron j is described by its weights m<sub>j</sub>. The weights of a cell translate to a value for each data type (e.g., bed roughness or magnetic anomaly), they can therefore also be understood as coordinates in the multidimensional data space.

For a given data point  $x_i$ , a best-matching neuron with the weights  $m_b$  is chosen in such a way that the Euclidean distance between  $x_i$  and  $m_b$  is minimized:

$$\|x_i - m_b\| = \min_i \{\|x_i - m_j\|\}$$
 (1)

Besides the weights, a neuron also has a location on the self-organizing map, which is described by the coordinate r in a two-dimensional space.

The network is trained iteratively t times for a randomly chosen input data point  $x_i$ . The best-matching neuron for this data point is determined, and then the weights of it and its neighbors are adjusted towards  $x_i$ . The value of the adjustment is determined by a neighborhood function  $h_{bj}(t)$ , it will be 1 for the best-matching neuron and decay as the neuron is further away from the best-matching neuron on the two-dimensional map. As a result of this neighborhood function, the map is trained so that neighboring cells on the map have similar weights and therefore will have similar data points mapping to the same cluster. Additionally, for convergence purposes, a time-dependent learning rate  $\alpha(t)$  is employed.

The training of a cell  $m_i(t)$  can be expressed as follows:


$$m_i(t+1) = m_i(t) + \alpha(t)h_{bi}(t)[x_i - m_i(t)]$$
 (2)

The choice of the neighborhood function can vary, and we utilize a Gaussian function:

$$h_{bj}(t) = \exp(-\frac{||r_b - r_j||}{2\sigma^2(t)})$$
 (3)

Here,  $r_b$  and  $r_j$  represent the locations of the best-matching neuron and the neuron to be trained on the self-organizing map, respectively. The parameter  $\sigma$  influences the smoothness of the computed map.

The two-dimensional SOM does not represent a geographic map; it is an arbitrary lower-dimensional representation of the higher-dimensional training dataset. E.g., an area with crystalline rocks with high gravity and magnetic anomaly values, as well as a rough bed, will appear close to similar areas, even though they are geographically far apart.

# 3 Data and analysis

#### 105 **3.1 Datasets**




We use the NASA Operation Ice Bridge (OIB) dataset collected between 2009 and 2012 (Figure 2) and high-level data products derived from this dataset. The radar data were recorded using the Hi-Capability Radar Sounder (HiCARS) Version 1 and later on Version 2 instrument. (MacGregor et al. 2021). We used the derived bed elevation from the radargrams (Blankenship et al. 2012, 2017). This dataset, however, includes several short-distance data gaps even in areas where bed echo is clearly visible in the radargram. This results in larger gaps in derived attributes, as observed in Eisen et al. (2020). We applied an optimization algorithm that filled each gap with sufficiently strong returns automatically. It specifically maximized the amplitude and the vertical gradient of the amplitude along the chosen bed elevation while minimizing the length of the bed elevation path (Liebsch 2023).

Figure 2: Data along the flight lines as input for the SOM analysis: A) Bed elevation from radar data (NASA Operation Ice Bridge), B) Magnetic anomaly (after Golynsky et al. 2018), C) Bouguer gravity anomaly (after Scheinert et al. 2016).

Magnetic data are taken from the ADMAP-2 compilation (Golynsky et al., 2018) along the OIB flight lines. In the supplementary database to Golynsky et al. (2018), the processed line data from the individual surveys are available, which are the basis for the ADMAP-2 map. Compared to the original flight data, the data are slightly smoothed, but suitable for our approach. For details on the magnetic processing, see Golynsky et al. (2006, 2018).

Gravity data were also collected as part of the OIB surveys. Unfortunately, the available gravity data have data gaps and only parts of the data is available in a pre-processed format (see coverage in MacGregor et al. 2021). Instead, we use the compilation from Scheinert et al. (2016). This 10-km-grid dataset has been sampled along the flight lines to treat it as survey data. Although, resampling cannot provide the full resolution of the survey data, we deem this adequate for our purpose, as the distance (height) between the point of observation (airplane) and the ice-bed interface is typically 3-5 km in the study area, leading to only minor loss of information when using the gridded gravity signal.

# 3.2 Attributes



We used the above datasets to generate 30 attributes (Table 1) for the SOM analysis. Several attributes were derived from a single dataset (e.g., bed elevation). Not only does the signal amplitude (e.g., bed elevation) characterize the signal, but also the spectral characteristics and local variations (e.g., roughness). This choice is subjective; therefore, rather than limiting the number of attributes, we include various attributes even though some attributes presumably have very similar characteristics and including both may have little impact on our result compared to including only one of them. Figure 3 shows some examples, while Figures A1-A3 show all attributes as normalized maps.

135 Figure 3: Example of attributes used for the SOM. A) Basal roughness ε derived from spectral domain b) Spectral Power in a 5-15 km wavelength bin from magnetic data and c) Shape index for gravity data. See text for more details and Figures A1-A3 for all individual attributes.

Attributes like roughness from radar data or spectral power in the short-wavelength magnetic field provide information about the variability in subglacial properties, e.g., a crystalline basement-ice interface can be expected to have a stronger contrast and larger variability than an incoherent bedrock (e.g., sedimentary basin) -ice interface. Other attributes based on the gravity and magnetic data (e.g., curvature) are well-suited to describe the changes between data points, while features like the shape index or the tilt derivative are also known to reflect the source characteristics. For some of these, Li (2015) provides a detailed analysis of the link between source geometry and observed field.

|                                            | Attribute                                                            | Abbreviation           |
|--------------------------------------------|----------------------------------------------------------------------|------------------------|
| Radar/Bed Elevation<br>(window size 10 km) | Isostatically adjusted bed elevation                                 | Isoadjusted topo       |
|                                            | Spectral Centroid Bed                                                | Centroid bed           |
|                                            | Roughness Attribute $\xi$ (500 m to 2000 m wavelength)               | $\xi$ bed              |
|                                            | Roughness Attribute $\eta$ (500 m to 2000 m wavelength)              | η bed                  |
|                                            | Variogram value (700 m to 800 m bin)                                 | v bed                  |
|                                            | Hurst coefficient (0 m to 1000 m)                                    | h bed                  |
|                                            | Moving average filtered bed elevation                                | Mean bed               |
|                                            | Standard deviation in a moving window                                | Stdev bed              |
|                                            | Kurtosis in a moving window                                          | Kurtosis bed           |
|                                            | Roughnes Attribute derived from the radar Tail $\sigma$ (this study) | σ                      |
| Magnetics<br>(window size 40 km)           | Magnetic Anomaly                                                     | mag                    |
|                                            | TDX Signal                                                           | TDX mag                |
|                                            | Spectral Centroid                                                    | Centroid mag           |
|                                            | Spectral Power in the 5 to 15 km bin                                 | Bin Power mag          |
|                                            | Moving average filtered magnetic anomaly                             | Mean mag               |
|                                            | Standard deviation in a moving window                                | Stdev mag              |
|                                            | Kurtosis in a moving window                                          | Kurtosis mag           |
|                                            | Curvature                                                            | Curvature mag          |
|                                            | Vertical gradient                                                    | VG mag                 |
|                                            | Analytical Signal                                                    | AS mag                 |
|                                            | Detrended Magnetic Anomaly                                           | Detrended mag          |
| Gravity<br>(no Windowed Attributes)        | Isostatic Anomaly                                                    | Iso grv                |
|                                            | Vertical gradient                                                    | VG grv                 |
|                                            | Analytical Signal                                                    | AS grv                 |
|                                            | TDX Signal                                                           | TDX grv                |
|                                            | Mean Curvature                                                       | K <sub>mean</sub> grv  |
|                                            | Gaussian Curvature                                                   | K <sub>Gauss</sub> grv |
|                                            | Min Curvature                                                        | K <sub>min</sub> grv   |
|                                            | Max Curvature                                                        | K <sub>max</sub> grv   |
|                                            | Shape Index                                                          | SI grv                 |

Table 1: List of all attributes used for the SOM and explained in the text. See examples in Figure 3. See Figure 4 for the correlation between the different attributes and Figure 5 shows the weights for the attributes.

# 3.2.1 Radar/Bed elevation attributes

In the following, we describe the 10 attributes based on bedrock elevation and radar data. For example, roughness can be computed in various ways from the bed elevation data. We used the same four roughness attributes as Eisen et al. (2020).

### 150 Isostatically adjusted bed elevation $t_{iso}$ (Isoadjusted topo)

As we are interested in local variations, we used the isostatically adjusted topography  $t_{iso}$ . This attribute is the hypothetical topographic height of the landscape assuming that no ice is present. In a simplified form, disregarding dynamic effects, it can be estimated from the ice surface height s and bed elevation z using the concept of isostasy after Airy with:

$$t_{iso} = (s - z) *917/3200 + z.$$
 (4)

#### 155 Spectral Centroid Bed (Centroid bed)

The spectral centroid represents the mean of all frequencies in the spectrum f(n), weighted by their spectral power S(n).

$$Centroid = \frac{\sum_{n=0}^{N-1} f(n) \cdot S(n)}{\sum_{n=0}^{N-1} S(n)}$$
 (5)

The Centroid indicates where the center of mass of the spectrum is located

Spectral Roughness Attribute (η bed)

 $\xi$  is the integrated power spectral density of the bed elevation profile in a 500 m to 2000 m wavelength bin. Given with the following equation:

$$\xi = \int_{k_1}^{k_2} S(k) \, \mathrm{d} \, k \tag{6}$$

where S is the power spectral density and k the wavenumber in the spectral domain.

Spectral Roughness (η bed)

To also capture horizontal changes in the spectral properties, Li et al. (2010) suggest including the integrated power spectral density of the horizontal derivative of the bed elevation  $\xi s_1$ , analogous to  $\xi$ . The spectral roughness attribute  $\eta$  is defined as:

$$\eta = \frac{\xi}{\xi_{sl}} \tag{7}$$

Variogram value (v bed)

This roughness attribute is derived from a variogram derived from a window along the flight line. We use a bin covering a 700 m to 800 m lag distance.

Hurst coefficient (h bed)

To complement the information on specific lag distances used in v, we also use the Hurst coefficient h. The Hurst exponent corresponds to the slope of the variogram in a log-log plot and can be described as:

$$v(\Delta x) = v(\Delta x_0) \left(\frac{\Delta x}{\Delta x_0}\right)^h \tag{8}$$

#### 175 Moving averaged filtered bed elevation (Mean bed)

To avoid using the bed elevations directly and reduce noise we used a 10 km-moving average filtered bed elevation

Standard deviation in a 10 km moving window (Stdev bed)

We compute the standard deviation of the bed elevation z in a 10 km moving window.

$$\sigma_{bed} = \sqrt{\frac{1}{N} \sum_{n=1}^{N} (z_i - \overline{z})^2}$$
(9)

where N is the number of points in a window. And  $\overline{z}$  is the mean of bed elevation in the window

Kurtosis in a 10 km moving window (Kurtosis bed)

Analogous to the standard deviation, the kurtosis w can be computed:

$$w = \frac{1}{N} \sum_{n=1}^{N} \left( \frac{z_i - \bar{z}}{\sigma} \right)^4 \tag{10}$$

*Bed Echo Tail Attribute* (σ)

We additionally derive an attribute from the shape of the bed echo. The direct interpretation of reflectivity can be challenging due to unknown attenuation within the ice (Matsuoka et al. 2011). Instead, we use the tail of the bed echo, which refers to the recorded signal after the initial backscattering from the bed has occurred. The tail originates from off-nadir backscattering. A significant advantage of this approach is that a radar ray scattered at the nadir and one scattered off-nadir encounter approximately the same conditions on their way back. Consequently, the shape of the bed echo tail can be described without relying on knowledge of attenuation.

To keep the fitting procedure stable and computationally efficient across the varying conditions of the survey area, we assume a simplistic Gaussian decay of the amplitude. This neglects losses due to beam characteristics and spherical spreading. The amplitude A as a function of incident angle  $\varphi$  is given as:

$$A(\varphi) = A_0 \cdot exp\left(\frac{-tan^2(\varphi)}{2\sigma^2}\right) \tag{11}$$

The bed echo tail  $\sigma$  can then be computed as the weighted average of  $tan(\varphi)$ :

$$\sigma = \frac{\sum_{i=1}^{N} A_i \cdot tan(\phi_i)}{\sum_{i=1}^{N} A_i} \tag{12}$$

#### 3.2.2 Magnetic data attributes

For the magnetic data, 11 attributes were computed along the flight lines (see Figure A2). Since there are some data gaps, certain attributes are also computed in the spectral domain, using a window with a length of 40 km. These attributes are standard features used to describe the magnetic field. See Blakely et al. (1996) or Li et al. (2015) for more details.

Magnetic anomaly (Mag)


This corresponds to the total field anomaly along the flight lines as explained above.

*Tilt Derivative (TDX mag)* 


The TDX signal is the tilt derivative of the magnetic field (Salem et al. 2008) computed as

$$TDX = \arctan(HG/Mzz)$$
 (13)

where HG is the total horizontal gradient and Mzz, the vertical gradient

Spectral Centroid (Centroid mag)

Typically, magnetic fields are inspected in a power spectrum to identify the source depth. Here, we calculate the spectral centroid of the power spectrum for a 40 km window using the following equation:

$$Centroid = \frac{\sum_{n=0}^{N-1} f(n)S(n)}{\sum_{n=0}^{N-1} S(n)}$$
 (14)

Hereby, the spectral centroid represents the mean of all frequencies f(n) in the spectrum, weighted by their spectral power S(n).

Spectral power bin (Bin Power mag)

The spectral power of the magnetic anomaly  $\varsigma_{mag}$ , limited to a bin of 5-15 km wavelength, is calculated using the following equation:

$$\varsigma_{mag} = \int_{5 \, m}^{15 \, km} S_{mag}(k) dk$$
 (15)

where  $S_{mag}$  is the classical power spectrum calculated in the wavenumber domain k. The range of 5-15 km has been chosen to represent subglacial sources, suppressing longer wavelengths due to regional sources and noise in the short-wavelength range. This attribute intends to represent the wavelength corresponding to the top bedrock and is shown as an example in Figure 3B.

Moving average filtered magnetic anomaly (Mean mag)

This was computed by removing a linear trend from the signal within a 40 km window around each point. This attribute is enhancing the short-wavelength content in the data.

Standard deviation in moving window (Stdev mag)

The attributes represent the variability of the signal in a 40 km window around each point. Details on the calculation are provided for bed/radar data above.

#### 225 Kurtosis in a moving window

Kurtosis is a measure to describe the sharpness of the magnetic anomaly. Details on the calculation are provided for the bed/radar data above.

Curvature (Curvature mag)

The curvature K is calculated along the flight line by

$$230 K = -M_{xz}/2M_z (16)$$

where  $M_{xz}$  is the gradient along the flight line (x-direction) of the vertical magnetic field component  $M_z$ .

More details on curvature calculations can be found in Li et al. (2015).

Vertical gradient (VG mag)

This is the vertical derivative of the vertical magnetic field component:

$$VG = M_{zz} = \frac{\partial Mz}{\partial z} \tag{17}$$

Analytical signal (AS mag)

The analytical signal is calculated from the vertical gradient and the gradient along the flight line as follows:

$$AS = \sqrt{M_{xz}^2 + M_{zz}^2} \tag{18}$$

Detrended Signal (Detrended mag)

The magnetic total field anomaly was detrended by removing a linear fit of the signal for a 40 km window around each data point. By removing such a linear trend, the attribute is more sensitive to local scale variations.

# 3.2.3 Gravity data attributes

For the gravity data, 9 attributes (see Figure A3) were computed from the grids, not along the flight lines. As we have gridded data, the derivatives are calculated using an equivalent source approach with prisms as source bodies. The prisms extend from the ice bed to a depth of 10 km. The densities of the prisms are estimated by inverting the gravity field of Scheinert et al. (2016). From these prisms, all spatial derivatives can be forward calculated following Nagy et al. (2000). For the curvature attributes, we are following Li et al. (2015), where the full mathematical background, tests with synthetic data, and an evaluation of these attributes for airborne gravity gradients can be found. See also Ebbing et al. (2018) for an example of curvature attributes from satellite gravity data over Antarctica.

Isostatic anomaly (Iso grv)

To obtain the isostatic anomaly, the free air anomaly was first mass corrected using the ice and bed elevation model Bed-Machine Antarctica v2 (Morlighem et al., 2020). To minimize isostatic effects, the undulation of the Moho boundary was estimated assuming Airy isostasy with an assumed density contrast of 530 kg/m<sup>3</sup> and a reference depth of 25 km. The resulting undulation was then forward modelled using prisms with the same density contrast and subtracted from the mass corrected anomaly.

Vertical gradient (VG grv)

The vertical gradient of the isostatic anomaly is calculated as

$$VG = G_{zz} = \frac{\partial lso \ grv}{\partial z} \tag{19}$$

Analytical signal (AS grv)

In contrast to the magnetic data, we calculate here the 3D analytical signal using

$$AS = \sqrt{G_{zx}^2 + G_{yz}^2 + G_{zz}^2} \tag{20}$$

where  $G_{xz}$ ,  $G_{yz}$  and  $G_{zz}$  are the derivatives in the x-, y- and z-direction of the isostatic anomaly, respectively.

Tilt derivative (TDX grv)

See description for attribute Tilt Derivative of the magnetic field (TDX mag).

Mean curvature ( $K_{mean}$  grv)

When curvature is used to interpret gravity anomalies, we try to delineate geometric information of subsurface structures from an observed non-geometric quantity. The mean curvature is calculated as

$$K_{mean} = \frac{G_{xx} + G_{yy}}{2G_z} \tag{21}$$

where  $G_{xx}$ ,  $G_{yy}$  are the second derivatives in the x-, y-direction.  $G_z$  is the isostatic anomaly.

Gaussian Curvature ( $K_{Gauss}$  grv)

The Gaussian curvature is the product of minimum and maximum curvatures and often exhibits rapid sign changes.

$$K_{Gauss} = -\frac{G_{xx}G_{yy} - G_{xy}^2}{G_Z^2} \tag{22}$$

*Maximum Curvature* ( $K_{max}$  grv)

From the two attributes before, we can calculate the maximum curvature:

$$K_{max} = K_{mean} + \sqrt{K_{mean}^2 - K_{Gauss}^2} \tag{23}$$

Minimum Curvature ( $K_{min}$  grv)

And similar as before, it follows the minimum curvature:

$$K_{min} = K_{mean} - \sqrt{K_{mean}^2 - K_{Gauss}^2} \tag{24}$$

Maximum and minimum curvature can be combined as well to compute the shape index.

$$SI = \left(\frac{2}{\pi}\right) \arctan\left[\left(K_{max} + K_{min}\right) / \left(K_{max} - K_{min}\right)\right] \tag{25}$$

The shape index is shown as an example for the gravity attributes in Figure 3c.

### 3.3 Calculation of the Self-Organizing Map

For the calculation of the SOMs, we use the existing Python package MiniSOM (Vettigli, 2018). Before training a SOM, all attributes are normalized using their standard deviation. Additionally, we removed all values deviating by more than ten standard deviations from the mean, as likely measurement errors. The threshold was arbitrarily chosen to exclude extreme outliers conservatively. All remaining points are part of the training data set. A unified distance matrix is computed that contains the distance to neighboring neurons for each neuron.

The resulting SOM has a shape of 30 by 30 and was trained using 15,000 iterations and a learning rate of  $10^{-4}$ .  $\sigma$  was set to 5 to create soft weight maps and avoid overfitting. Naturally, there are numerous possibilities and parameter sets that yield acceptable results. For visual comparison only, the final map was divided into 5 clusters, where the main attributes show similar values. Boundaries were chosen in a way that neighboring cells are distinct from each other.

#### 4 Results and discussion


#### 295 **4.1 Correlation between attributes**

We first examined correlations between individual attributes (Figure 4). Particularly high correlations or anticorrelations indicate how different datasets are affecting each other, and which ones can be used jointly in an interpretation. The correlation matrix between the attributes shows that, in general, the correlation is strongest between attributes derived from the same data type (radar, magnetic, or gravity), as expected. Some of the attributes do not follow this general observation. E.g., the Tilt-Derivative of the gravity (TDX grv) correlates stronger with radar roughness than with any other gravity-derived attribute. Roughness reflects, first of all, variations in the topography itself. Such a varying topography will cause variations in the gravity signal and, to a minor extent, the magnetic signal. This is evident in the correlations between roughness and spectral attributes in magnetics, as well as with the gravity signal, which may indicate that a smooth bed-ice transition tends to be less dense and has lower susceptibility.

Some of the attributes show almost no correlation with other attributes, such as Tilt-Derivative of the magnetic field (TDX\_mag) or Gaussian Curvature of the gravity field (K<sub>Gauss</sub>). An absence of correlation might indicate that these attributes are sensitive to different source structures.

Another example is the correlation of the Total Magnetic Field anomaly (Mag) and its detrended version (Detrended mag). While the first shows some degree of correlation to the gravity-derived attributes, the second does not. That corresponds to the

different sensitivity of the gravity and magnetic field to the sources, but might also indicate that we miss some of the gravity signal by using a gridded data set as input and not measurements along the flight lines.

Other attributes, such as the roughness attributes (ζ bed, η bed), show a correlation with both gravity and magnetic attributes, for example, the spectral centroid (Centroid mag) or the shape index (SI\_grv). Similarly, the power of the 5 to 15 km bin (Bin Power mag) correlates with the basal roughness attributes. This could indicate that sedimentary basins lack short-wavelength signals as they tend to have smoother surfaces. Similarly, correlations between the gravity attributes could support the idea that dense rocks tend to be more erosion-resistant, leading to rougher landscapes.

Figure 4: Correlation matrix for all attributes listed in Table 1.

#### 4.2 Weights for individual attributes


An important parameter to consider is the weights for the individual attributes (see Figure 5) in contributing to the chosen SOM (Figure 6). The SOM is not a unique solution as it depends on the attributes as well as on the initialization and chosen thresholds. Hence, even with the same choice of parameters, the outcome may vary and any SOM must be considered as only

a possible solution. If weights are near zero across the whole map for a specific attribute, that indicates that the attribute has no significant impact on the SOM and could be omitted from the analysis without significant loss of information. The weights map shows that some of the attributes, e.g., SI\_Grav, strongly influence the results, while others, e.g., Kurtosis mag and bed, have a minor impact. That corresponds to the correlation with other attributes (Figure 4). Those attributes not correlating with other attributes have in general, less impact on the final SOMs, while those showing a larger degree of correlation are deemed more important. That must be taken into consideration when discussing the dependency of the final SOM on the choice of attributes for analysis.

Figure 5: Weights for every attribute and cell (30 by 30) of the SOM. All attributes were rescaled using the standard deviation, before the training started. See Figures 3 and A1-A3 for a geographical representation of the individual attributes.

#### 335 **4.3 Subglacial clusters from SOMs**



Next, we analyze the SOMs in more detail by discussing apparent clusters in the map. For a first comparison between our SOM and the bed type classification by Aitken et al. (2023), we map their classification on our 2D representation (Figure 6).

Figure 6: Visualization of the SOM and class distribution. Every data point (measurement along a flight line) was assigned a class according to Aitken et al. (2023) and subsequently mapped onto the SOM. Each cell represents a neuron in the SOM and contains the data points mapped to it. The pie charts within each cell indicate the proportions of different classes present. The letters A–E highlight regions of the SOM with similar properties; they are manually defined to aid description and interpretation.


The crystalline-basement class indicates where the bed is interpreted to consist of igneous or metamorphic rocks (including high-grade metasedimentary rocks), with either no or only a thin veneer of sedimentary cover. Typically, these regions possess the characteristics of high elevation and high gravity with high spatial variability in topography, gravity, and magnetic data. Type 1 basin class represents regions where sedimentary basins are preserved in relatively unmodified basins, with typical characteristics of low elevation and low gravity, and low spatial variability in gravity and magnetic data. Along-track roughness tends to be low. The intra-basin volcanics class includes areas where volcanic rocks are interpreted to be emplaced within a Type 1 basin sequence. Type 2 basin class, in turn, represents areas where sedimentary rocks are known or inferred but the original depositional basin is not preserved. These rocks tend to predate the formation of the present landscape, are often uplifted to high elevations, may be intruded by younger igneous rocks, may be heavily eroded and may have geophysical characteristics more similar to crystalline basement than Type 1 basins. For mixed classes the geophysical characteristics are not providing clear evidence for an assignment to a single class (Aitken et al. 2023). We expect our SOM to contribute the

most to an improved understanding of the mixed or inconclusive classes. For comparison, we sample for each data point of our SOM its class according to Aitken et al. (2023) shown in Figure 6. The datapoint can then be mapped into the two dimensions of the SOM. The pie chart for each cell of our map represents the different classes mapped to it, e.g., association with Type 1 basins and Crystalline Basement is dominant on the left and right side, respectively, while most cells sample different domains and can be less clearly associated with a certain class. As the SOM is mapping data firstly in a 2D Domain based on attribute similarity and irrespective of the geographic location (see inset in Figure 7), the domains A to E seen in Figure 6 and Figure 7 are only to guide visual comparison and are not based on a statistical evaluation of the results.

Type 1 basins are predominantly located within cluster C, aligning with the expected characteristics of smooth beds, low gravity, and minimal magnetic signals. In contrast, crystalline rocks are predominantly found in cluster E. This observation supports the assumption that strong magnetic anomalies are typically generated by crystalline rocks. Furthermore, crystalline rocks are also seen on the left side of cluster B. This sub-cluster exhibits high roughness and intense magnetic and gravity signals, as expected for crystalline rocks. Type 2 basins, however, do not show a distinct concentration but are visible across various regions of the map. This dispersion raises questions about the feasibility of correctly inferring this class solely from the attribute compilation used here or from the robustness of defining this class over such a large region. Possibly, the Type 2 basins, in this region, mainly sedimentary rocks on highlands, have a more heterogeneous build-up or reflect different subtypes compared to the interpretation by Aitken et al. (2023). For the mixed class, no clear domain can be found on the SOM, conforming to their complex nature.

Figure 7: Representation of the clusters from the SOM. A) 2D Colormap for the SOM. B) Unified distance matrix (30 by 30 cells) for the presented SOM. C) Geographical distribution of the SOM. The orange box indicates the zoom-in area in Figure 8.

In Figure 7 and 8, we map the SOM along the individual flight lines for geographical representation, and in Figure 9 along a profile.



Comparison with the bed type classification of Aitken et al. (2023) shows a general agreement (Figure 8). Particularly, the delineation of various highlands corresponds closely between the two classifications. However, for some structures, as Knox Highlands (classified as Crystalline) and Highlands A (classified as Type 2 Basin), there are differences in the results. This is consistent with the observation that the Type 2 Basin class seems to be mapped for quite dissimilar physical settings.

Additionally, most basins, including the Southern Wilkes Basin, Central Aurora Basin, and Aurora South Basin, exhibit strong consistency with the classification presented by Aitken et al. (2023). Furthermore, the sedimentary basin likelihood map as presented by Li et al. (2022) consistently indicates thick sedimentary layers in areas that were mapped within class C. The most significant disagreement between the SOM and the classification by Aitken et al. (2023) is shown for the Sabrina Basin and Aurora North Basin. In these areas, the fine-scale variations within Clusters A, D, and E of the mapped SOM appear to

contradict the homogeneous classification by Aitken et al. (2023), suggesting that the SOM may be able to capture local variations, which are best observed when compared along an individual flight line (Figure 9).

Figure 8: Zoom in for the Wilkes land area for comparison of our results and Aitken et al. (2023): (A) Geographical distribution of the SOM, (B) Geological bed types from the study by Aitken et al. (2023). See Figure 7 for location and orange line indicates profile shown in Figure 9.




Along a flight line (Figure 9), the interpretation by Aitken et al. (2023) does not clearly follow the boundaries visible in the data and SOM. The radar data show that there are sections of the basin where no return from the bed was detected (e.g., distance ~100 km), while it appears as a very smooth reflector in places where it was detected (~140 km). Additionally, the magnetic signal exhibits a predominantly long wavelength above the basin and shows no obvious correlation with the bed. These observations indicate the presence of non-magnetic rocks near the bed. The SOM effectively captures the abrupt change at the rise of Aurora North Basin in the north of the profile (Figure 9). For clusters B and E, the correlation between the magnetic signal and bed elevation becomes evident. This suggests the presence of magnetic rocks near the surface of Aurora North Basin, whereas it is not the case in the Aurora Basin. This illustrates how the SOM can successfully integrate information from various data types into a single parameter, clearly highlighting the most probable geological boundaries. It therefore

could be a useful tool for future mapping attempts and could also help adjust boundaries while leveraging all available data types.

Figure 9: A combined plot of magnetics, gravity and radar data along a profile. Beneath the plots, the SOM is shown along with the classification by Aitken et al., 2023. Color coding for SOMS is according to Figure 7A.

# 4.4 Pitfalls and possibilities of SOMs




The comparison to the expert judgment approach by Aitken et al. (2023) by compiling available data sources, shows that SOMs can potentially provide an added level of detail or aid in detecting possible errors or inconsistencies as it should be based on measured data as much as possible. Nevertheless, the application of SOM has certain limitations. For example, the result is based on the selection of attributes. Some attributes show generally low correlation with other attributes (Figure 4) and might be omitted (e.g., detrended magnetic field). For others, like the isostatic gravity field and its vertical gradient seems to add little additional information and only one may be used. Instead, additional attributes derived either from the data used here or other independent datasets (e.g., roughness derived from ice surface elevation or ice flow velocity) could be added.

As such data sets often have quite dissimilar coverage, we limit ourselves here to use data sets with a similar coverage and sampling. The number of attributes is intended to avoid a bias towards a single data set. However, we did not test how the results would vary using a different number or only a selection of the attributes, mainly due to computational reasons, but also due to the different characteristics of the input data set (flight lines and resampled gridded products).

Using a consistent data set, e.g., data based on the same flight campaign, is preferred to improve interpretation at survey scale, with the trade-off that along-line variations may be underestimated if line orientations are not located perpendicular to the main strike direction. But this trade-off appears to be preferential to the use of gridded data products, where interpolation and the lower resolution of grids compared to line products affect the quality of the resulting products. Still, the insensitivity to spatial anisotropy of gridded data products might outweigh the gains in data resolution. Furthermore, there are additional attributes that could be derived from gridded datasets, potentially enhancing the resulting SOMs. Certainly, exploring the choice of input datasets by assessing the importance of different attributes, possibly by jack-knifing, is worth exploring in more detail.

Another point of caution is that there is little control over the meaning of the output clusters of the SOMs. That implies that an interpretation is needed to assign meaning to each cluster. Here, as for other machine learning methods, a training or validation data set might increase the confidence in the results. Furthermore, not all features mapped by the SOMs might be a geological signal, but some, especially local features, might reflect data quality (measurement errors or noise). To generate a well-informed classification, multiple, quality-controlled, data types should be combined, and a careful assessment of the data products is required.

Despite these shortcomings, SOMs can aid in defining (geological) units with distinct properties and to help interpreters to make data-optimized classifications. Especially at the spatial scale that seems to be most important for understanding the coupling of ice-sheets and the underlying solid earth structure (e.g. McCormack et al. 2022), the SOMs can provide a second level of detail that is difficult to achieve from direct interpretation. This can be potentially combined with statistical analysis of bedrock properties from petrophysical samples to predict the variations of thermal parameters on a local scale (e.g., Freienstein et al., 2025). As always, careful evaluation of the final results is still a crucial point in estimating subglacial properties as the SOMs do not provide immediately a new geological map, but a tool for classification and interpretation.

#### **5 Conclusions**






We present a novel mapping of subglacial geology using Self-Organizing Maps applied to radar, gravity and magnetic data sets along flight lines from the NASA Operation Ice Bridge (OIB) dataset in East Antarctica. The attributes calculated from the data sets provide a suite of products useful for interpretation, however, challenging for direct manual interpretation. Here, the SOM helps to group the complex features into a simple map. Comparison to the classification of Aitken et al. (2023) generally shows good agreement for the major classes in regions of low complexity, while also indicating variations within some of the classes. In such areas, the SOMs can help to refine interpretations and unveil previously unknown small-scale structures. To further enhance the clustering capabilities of the SOM, an in-depth exploration of hyperparameters and choice of attributes could lead to improved results.

In general, data selection is a key to avoid a bias by inconsistent data sets and for example, the recently released geophysical data catalogue from the British Antarctic Survey includes multiple surveys with magnetic, gravity, and radar data (Frémand et al., 2022), presenting an opportunity to further explore the possibilities of SOMs for flight line data.

We see two possible directions as next steps. 1) The classification of different bed types could serve as a constraint for (joint) inversion, extending this analysis from a description of subglacial properties to a physical earth model, needed to describe the full coupling between the Solid Earth and the overlying ice-sheets. 2) An analysis of ice-sheet modelling to the complexity of subglacial geology, to identify which parameters and spatial-scales are most critical to predict the future evolution of the Polar ice-sheets.


**Appendices**: As an appendix, we present all 30 attributes in a normalized form as used as input calculating the SOM discussed and presented in Figures 5-9.

Figure A1: Normalized attributes based on topography and radar data. See text and Table 1 for more details.

Figure A2:. Normalized attributes based on magnetic data. See text and Table 1 for more details.

Figure A3. Normalized attributes based on gravity data. See text and Table 1 for more details.

**Code Availability:** Notebook and training data sets are available through Zenodo: https://doi.org/10.5281/zenodo.17250123 **Author contribution**: JL carried out the formal analysis and prepared all figures. JL and JE wrote the manuscript draft; KM reviewed and edited the manuscript; JE and KM supervised the study as part of the Master thesis by JL.

**Competing interests**: The authors declare that they have no conflict of interest.


Acknowledgments: JL acknowledges funding received by the Erasmus+ program of the European Union which made his stay at the Norwegian Polar Institute in Tromsø possible. We thank the editor Nicolas Gillet and the reviewers Tobias Stål and Fausto Ferraccioli for their thorough reviews and Alan Aitken for discussions on an earlier version, which helped us to improve our manuscript.

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
