# Peer review of "Application of Self-Organizing Maps to characterize subglacial bedrock properties based on gravity, magnetic and radar data – An example for the Wilkes and Aurora Subglacial Basin region, East Antarctica"

_EGUsphere, 2025_

## Referee Comment (RC2)

**TITLE**- I find the current title a bit misleading and would therefore recommend changing it (see below)

Application of Self-Organizing Maps to characterize subglacial  bedrock properties in East Antarctica based on gravity, magnetic and  radar data

the paper does not give an overview or a critical discussion of bedrock properties for whole of East Antarctica; instead it essentially presents a new self-organising map derived from multiple existing datasets and their derivatives that potentially (although unfortunately not really discussed in the current MS) may provide a basis for new interpretations of subglacial bed geology, compared to the relatively simple classification of the study region in sedimentary basins (Type I and 2), crystalline basement and mixed bed previously proposed by Aitken et al. (2023) in their continent scale Antarctic analysis.

**Suggested revised title**

**Self-organizing map of the Wilkes and Aurora Subglacial Basin region in East Antarctica based on radar, aeromagnetic and aerogravity attributes**

**Abstract-**

"Therefore, it is crucial to efficiently combine various attributes derived from satellite and airborne geophysical surveys to characterize subglacial properties"

**I find this claim difficult to follow**- what subgacial properties can we infer from satellites presently?-

 I would argue not many!

We don't have high resolution views neither of bed topography nor of magnetic or gravity patterns from satellite data to image the heterogeneity of bed properties - but we do have a good large-scale view of the crust and lithosphere especially from GOCE (e.g. Ebbing et al., 2018; Pappa et al., 2019a,b).

As this study does not deal with satellite data, I would suggest either dropping this sentence in the abstract -or state that while satellite data provides views of spatial and temporal changes in ice sheet dynamics, as well as glimpses into the deeper crust and lithosphere, airborne geophysical data are a key to resolve the heterogeneity in subglacial geology and its properties.

"we evaluate a Self-Organizing Map (SOM), an unsupervised machine learning technique. The concept of SOMs, an unsupervised machine learning approach". **There is repetition here- please mention just once**

"for the case at hand" **This could make a potential reader wonder other than a methodological application what the broader purpose of the study itself is.**

**Instead, I would strongly recommend highlighting earlier on in the abstract (right at the beginning) the key importance and impact of the study area**- aka the Wilkes and Aurora Subglacial basins are huge marine-based sectors of the EAIS that could potentially be prone to dynamic instability in the future. Hence understanding the influence of the heterogeneity in subglacial geology and bed topography is particularly important here.

Then I would say that the interpretation of these critical regions has so far relied on expert interpretations of airborne geophysical data but here you explore the application of a less subjective self-organising map as a tool to xxx etc.

"where also suitable data sets for the application of the SOM exist"

I would drop this sentence- we actually have many other areas in both East and West Antarctica where we have considerably better datasets than the ICECAP data used herein, which is mostly composed of very widely spaced radial lines (which is certainly NOT ideal for resolving the fine scale heterogeneity in subglacial geology and bed properties from aeromagnetic data for example- although it does give a large scale perpective of these huge marine based sectors of the EAIS that could not be achieved without a huge logistical effort if the surveys had been flown on conventional regularly spaced surveys such as those flown over the northern Wilkes Subglacial Basin (see also Ferraccioli et al., 2007, Terra Antartica Reports);

Incidentally it is NOT just NASA data. The data you are using are from a major joint multinational US-UK-Australian and French and Italian effort ICECAP that was also BUT not exclusively supported by NASA! The data may have been downloaded from there but full recognition to ICECAP should also be given in the text.

"Previous analysis indicated the presence of both crystalline basement and sedimentary basins in the area, and our SOM shows a remarkable agreement"

**I would recommend rewording this**- highlight first what your approach brings to the table compared to the current state of the art- e.g. it unveils additional heterogeneity in the characterisation of cryptic properties of the subglacial highlands compared to continental-scale maps and then state that it aids depicting the distribution of sedimentary basins and crystalline basement domains in general agreement with previous expert interpretations based on aerogeophysical datasets;

**incidentally what is new from this SOM map is actually pretty difficult to follow also because the lack of a sufficiently well-founded geological or geomorphological interpretation and discussion sections in the current MS.**

"These variations can potentially be exploited further in describing subglacial properties and the coupling between bed and overlying ice-sheets"-

**this raises the question of why this was not done here**.

I would drop this sentence in the abstract and perhaps leave this pointer pending for the discussion or conclusion.

**Unless the paper is re-written very extensively, I would suggest you focus here primarily on the new method and the map you deliver as a new tool mainly.**

**Introduction**

Aitken et al. 2023, Bell et al. 2008, McCormack et al. 2022. I would suggest adding

Bingham et al., 2012, *Nature* & Jordan et al., 2023, *Science Advances* to the list here

"airborne magnetic or gravity datasets". **I would prefer this to be reworded** into aeromagnetic and airborne gravity datasets.

Ferraccioli et al 2002, 2009

**I would recommend adding also** Ferraccioli et al., 2011, *Nature*;

**I would recommend adding also** Forsberg et al., 2018, doi: https://doi.org/10.1144/SP461.17

That presents data from another critical marine-based sector of the EAIS;

McLean et al. 2009, Aitken et al. 2014, Kim et al. 2022

- Kim's paper il more about a compilation of magnetic data using satellite data- I would drop here.

"the interpretation requires some form of constraint to overcome the inherent ambiguities"

- this is of course true wrt to aeromagnetic data interpretation -but I would argue this also holds true if one uses multiple methods and their derivatives too.

For example, basal roughness varies at different scales and is not only due to subglacial geology, although at large spatial scales there will obviously tend to me a rougher bed over a crystalline-basement dominated Precambrian craton that a major sedimentary basin.

For the devil in the detail- see for example Jordan et al., 2010 Terra Nova. They show a nice example of an area with apparently quite uniform subglacial geology (at least according to potential field data interpretations) but remarkable differences in basal roughness (speculatively linked to a huge paleo-subglacial lake and associated drainage system);

Aeromagnetic and airborne gravity data may not resolve thin sedimentary drapes in marine basins (that is basins where marine incursion occurred in response to major ice sheet retreat atop of the pre-existing major sedimentary basins) at all;

Conversely, these data are more likely to image thick sedimentary basins that generally predate the EAIS by hundreds of millions of years- e.g. the Devonian to Jurassic Beacon Superbasin in the Wilkes Subglacial Basin region (see Ferraccioli et al., 2009a).

Where we have exposures of the rocks we can directly see that the mesa topography associated with Beacon Supergroup rocks massively intruded by Jurassic Ferrar sills tends to divert ice flow around it. Hence the patterns in the roughness of the bed are significantly more complex than a simple "smooth bed" atop of a sedimentary basin.

Overall, when one combines different methods it is important to address also how the individual limitations then combine together, also as a function of the features that one is then aiming to interpret in the first place (e.g. sedimentary basins, crystalline rocks-dominated mountain ranges etc). And of course, one has to be careful-  there are huge mountain ranges dominated by sediments too that could have been incised by rivers and early glacial systems creating rough topography in highlands- see for example Rose et al., 2013 EPSL. A clear example of this are the Vostok Subglacial Highlands.

"data sets" change to datasets

"it is in part subject to interpretations"- well I would say that it is an interpretation!

It should be clearer in the text that Alan's map is an inferred subglacial geology map showing the proposed distribution of sedimentary basins (Type 1 and 2 as Alan called them... a terminology that notably I still find a bit cryptic as a geologist), mixed bed and crystalline rocks-dominated bedrock

"Please see Aitken et al. (2023) for more details on this".

I would not recommend using this type of sentence. Your paper needs to stand alone- and hence the main concepts and findings should be reviewed here- also because it is important to state more clearly what your own research objective is in the Wilkes and Aurora Subglacial Basin region. **Are you going to try and test and augment some of these previous interpretations with the aid a new tool?**

"Machine learning and statistical based methods are nowadays popular approaches for less heterogenous models". **This sentence is rather unclear to me**. What are you trying to say exactly here? That we don't use machine learning for inferring subglacial geology or something else?

**Self-Organizing Maps**- a bit more references here than a single reference from 1990! would have been helpful. Please provide more updated references.

"The study area is in Wilkes Land, East Antarctica (Figure 1), chosen for the excellent coverage with line data"

**Firstly**, this is NOT only Wilkes Land. The area you covering includes Victoria Land, George V Land and Wilkes Land

**Secondly**, there is NOT excellent aeromagnetic or aeromagnetic data coverage here.

**Instead, there are widely spaced reconnaissance ICECAP lines in the study region compared say to the many GITARA aeromagnetic surveys over Vitoria Land (see e.g. Ferraccioli & Bozzo, 2003- https://doi.org/10.1144/GSL.SP.2003.210.01.07 or Ferraccioli et al., 2009b- https://doi.org/10.1016/j.tecto.2008.11.028) or even compared to the WISE-ISODYN survey over the northern Wilkes Subglacial Basin** (Ferraccioli et al., 2009; Jordan et al., 2013; https://doi.org/10.1016/j.tecto.2012.06.041), **despite the key importance of this study region to comprehend the potential influence of heterogeneous bed topography properties on the behaviour of the marine based and hence potentially more vulnerable Wilkes and Aurora subglacial basin sectors of the EAIS.**

What is important to state (and needs to be added in the text) is that despite the large coverage in terms of area the line spacing of ICECAP surveys is wide especially in the interior of the EAIS (where by the way the onset of ice streaming occurs);

but your **SOM approach that exploits the full potential of line data still adds new views of the region when compared to interpretation methods based on grids alone**

 Interested reader- I think a little bit more background would have helped here;

**3.1 Study area**

**I find this is a rather incomplete description of the study area.**

**The various bedrock features, basins, and geological domains should come across much more clearly here, especially if one then wants to highlight later on that the SOM approach provides hints into further complexities than previously inferred.**

**And as noted above -the coverage is reconnaissance in character with very widely spaced lines especially in the interior of the EAIS due to the radial flight pattern of ICECAP**

 "consequently its massive potential for sea-level rise" – You need some references here!

(Fretwell et al., 2013)- is clearly NOT a good reference for the hot debate on the stability of this part of the EAIS. Please do the relevant literature research and cite the relevant papers.

And one needs to be updated to also add BEDMAP 3 that incorporates all the new data in this region! (references Pritchard et al., 2025, https://doi.org/10.1038/s41597-025-04672-y; Fremand et al., 2023, https://doi.org/10.5194/essd-15-2695-2023) including in the figure.

**3.2. Datasets**

We use the NASA Operation Ice Bridge (OIB) dataset collected between 2009 and 2012- ICECAP needs citing too!

Magnetic anomaly (taken from ADMAP-2 (Golynsky et al. 2018)?

**I am bit puzzled here**- you must have used a subset of ADMAP 2 here as we have a lot more aeromagnetic data in the study region- I presume you only used the re-processed ICECAP radial profiles we then compiled in ADMAP 2.

I suggest you use a transparent backdrop for all the datasets with BEDMAP 3, ADMAP 2 and the AntGG compilation and show the ICECAP radial lines you are re-analysing atop of the backdrop.

Effectively, despite all these additional data that would clearly help contextualise the geological and bed interpretation processes along the profiles you are going back solely to the individual ICECAP profiles which I find makes the paper mostly methodological focussed

*Figure 2.* The reversed magnetic anomaly scale with blue tones over highs and browns over lows is rather unconventional. I find it puzzling when compared to the original ADMAP 2 data or the input data of Aitken et al., 2014. Please change this!

Also, I wonder why the authors are not using the Ebbing et al., 2021 product that also replaces the long-wavelength component of the magnetic field with SWARM satellite magnetic data. By the way ICECAP data did not match very well higher resolution and better levelled data on the TAM side.

For the gravity data- I find the inverted colour scale with gravity lows shown in red confusing. Please change this!

**Airborne gravity data;** MacGregor et al. 2021 etc

If you asked NASA and the ICECAP team they would likely have given you the data. Also one has to be careful here. We did not have as good bedrock topography data compared to BEDMAP 3 at the time that we made the AntGG compilation in 2014 and then published it in 2016 in the Scheinert et al., 2016 GRL paper so ideally one would have:

1. Accessed the complete Free-Air gravity data
2. Recomputed the Bouguer anomaly based on the updated BEDMAP 3 compilation

Anyhow, if this is not possible at this stage to do this extra work, at least some sentence on the availability of more updated bed topography and ice thickness datasets in particular for Bouguer anomaly calculation should be made

"Instead of using the bedrock topography, it can be useful to determine the isostatic adjusted topography tiso".

**I disagree it is not instead- BOTH the original bed and a rebounded version of the bed should be used and discussed**.

That said the bed is NOT going to rebound in an Airy way when the EAIS is removed (that is essentially prior to its formation -as the EAIS is unlikely to have ever completely collapsed since its formation) or retreated –

because these are typically more complex visco-plastic process than predicted by an Airy model. Some explanation on the simplifications associated with the Airy model should be introduced. And one should explain for the non-expert reader that the Bouguer anomaly map is strongly influenced by crustal thickness variations and hence the use of Airy isostatic maps can help enhance the shallow level intra-crustal sources of the anomalies.

In addition, if one wants to make more meaningful comparisons with **pre-glacial topography and geology then one should consider that huge amounts of glacial erosion and sediment unloading**

**and flexure have occurred particularly in the region of these subglacial basins since EAIS initiation at the Eocene-Oligocene boundary –**

**so an interesting product to use would have been to also use the**

Paxman et al 2019, 2020 **paleotopography datasets** (https://doi.org/10.1016/j.palaeo.2019.109346; https://doi.org/10.1029/2020GL090003) **as this provides a more realistic view of the pre-glacial landscape and its links with pre-glacial geological boundary conditions**

**Attributes**

When you refer to the attributes please refer to the table upfront.

I would much have preferred to see some more profile views (e.g. at least a couple per basin and flanking mountain ranges) of the major features with multiple individual attributes displayed first and tied much better to a map view with a clearer overlay of the major subglacial basins and highlands and geological domains that one is investigating.

This is essential for any interpretation and discussion of the features.

Could be in the interpretation section but should be somewhere.

*Figure 3 Example of attributes used for the SOM*. The logic of this example is unclear to me. Why these rather than many others?

**B Spectral Power-** is this not influenced by distance to source?- the pink colours seem to correspond to outcropping Precambrian crystalline basement rocks

**3C** I find again the inverted colour scale un-conventional and the meaning of the shape index is also unclear to me

I believe it would have been better to present maps for related quantities fist, aka bed, magnetic and gravity followed by mixed attributes for potential correlations (or lack thereof).

**Some explanation of why the spectral centroid bed is used would be useful- what can this yield?**

ξ is the integrated power spectral density of the bed elevation profile in a 500 m to 2000 m wavelength bin.in a 500 m to 2000 m wavelength bin.

**Why was this wavelength used? Is there a wavelength dependency and what can different wavelengths be representative of?**

We use a bin covering 700 m to 800 m lag distance.  **Why and what impact does this choice have?**

we used a 10 km-moving average filtered bed elevation. **Why was 10 km chosen?**

The TDX signal is the tilt derivative of the magnetic field. **Ok but what effect does the tilt derivative have on the aeromagnetic data**- and what was the purpose of this enhancement?

I assume the aim was enhancing the contribution of shallower magnetic anomaly sources as a way of helping in defining the heterogeneity in the bedrock geology? Needs explaining for the non-expert reader.

*Moving average filtered magnetic anomaly (Mean mag)*

Enhancing the short wavelength content- explain why this is useful

*Analytical signal (AS mag)*

When using higher order derivatives a careful analysis of the noise should be provided. Even where much higher quality aeromagnetic data on regular grids are available e.g over northern Victoria Land (e.g. Ferraccioli et al., 2009b) we generally had to filter the data with Hanning or other filters so as not to have too much noise in AS products.

*Isostatic anomaly (Iso grv)*

I would have preferred the use of BEDMAP 3 that is based on line data as opposed to a product that includes ice sheet mass conservation such as BedMachine. Additionally, the density contrast used seems very high for East Antarctica 530 kg/m3.

*Analytical signal (AS grv)*

Again- with the higher order derivatives it is important to check that the noise levels are not rendering noise in the grid only. Also, it is important to note that the definition of out of plane bedrock topography and ice thickness outside the widely spaced survey lines is poor and hence in the Bouguer anomaly data itself there can already be quite a lot of noise that can tend to blow up with the high order derivatives. Some smoothing/filtering is recommended

**3.4 SOM calculation, used algorithms and other considerations**

Please provide a simpler and less vague (..other considerations) heading

e.g. SOM calculations is fine where limitations etc can be addressed in the text perhaps as a sub paragraph. The resulting SOM has a shape of 30 by 30- what?

**4 Results and discussion**

I would have preferred to have seen a more critical and focussed discussion presented and preceded by an interpretation

"Such a varying topography will cause a varying gravity and to a minor portion magnetic signal".

**I am not sure I understand this statement.**

High frequency magnetic anomalies will correlate with rough crystalline basement topography. I don't understand why this would be considered surprising. Say you go over northern Victoria Land where you see the rough bed associated with relatively well exposed granitoids and metamorphic rocks and you clearly appreciate in tilt derivative maps the high frequency anomalies that are caused by these rocks. There are of course some exceptions where the rock assemblages are magnetite-poor but at large-scale this holds.

"That corresponds to the different sensitivity of the gravity and magnetic field to the sources, but might also indicate that we miss some of the gravity signal by using a gridded data set as input and not measurements along the flight lines"

That is not likely to be the only reason. It is important to note that airborne gravity data are heavily filtered typically with a ca 9 km half width filters (see e.g. Jordan et al., 2013- https://doi.org/10.1016/j.tecto.2012.06.041)

4.2 Weights for individual attributes

Before we analyse the SOM in more details with respect to its possible (geological meaning) please take the parenthesis out of the sentence does not make sense

*Figure 5 Weights for every attribute and cell of the SOM. All attributes where rescaled using the standard deviation, before the training started*.

I am not sure I understand this figure.

What are we seeing the whole study area domain or something else?

**Figure 6 Visualization of the SOM and class distribution**

I find this figure quite difficult to follow.

Before having everything displayed together it may perhaps have been clearer to see each category of features proposed by Aitken et al., 2023 displayed individually (and ideally also with selected examples extracted in the correct geographic location) vs the SOM.

Aka to what extent can one define for different areas Type 1 basins, Type 2 basins, mixed beds etc using a SOM approach? (also considering large differences between the widely spaced interior regions and the coastal sectors where coverage is much better).

I am also struggling a lot to have a spatial view. I assume that although we are seeing a map this is not locating the corresponding features in map view.

I also don't understand the letters and the dashed lines that well. For example, when I look at region E there seems to be a remarkable contrast between the red and blue regions with little similarity between them -so I fail to understand while this all labelled as a single feature E.

*Figure 7: Representation of the clusters from the SOM. A) Geographical distribution of the SOM., B) Unified distance matrix for the presented SOM. C) Classification of geological bed type from Aitken et al. (2023). Yellow line indicates profile of Figure 8*.

**This figure is difficult to follow. Firstly, the colour scale is very difficult to follow as the differences between the colours between A and C and D and E is hard to appreciate. I am not sure the text has explained clearly enough what the unified distance matrix shown in panel b is and panel c has no km scale or coordinate label. It is critical that you show the previous interpretation of Aitken et al., 2023 with labels of the geographic features you then refer to in the text at the SAME scale as the colour map of the SOM. Furthermore, I would suggest that you show an additional panel with these overlain so one can start thinking about the potential interpretation of additional heterogeneities not addressed in the continental scale interpretation of Aitken et al.  Also where is the yellow line showing the location of Figure 8?**

**Figure 8 should also report locations. What basin and highlands are we looking at?**

I would have preferred at least a couple of profiles per major sublglacial basin and a more detailed discussion of results -not just the methodological part.

---

## Author Comment (AC1)

**Response to detailed comments on the MS by Liebsch et al. by Fausto Ferraccioli**

We thank the reviewer for his insightful and detailed comments. We largely followed your advice. However, we do not as suggested, put more emphasis on the geological interpretation, but, as advised by the second reviewer Tobias Stål, on presenting the possibilities on using SOMs from a methodological side.

Detailed geological interpretation goes beyond our initial intention and certainly would require adding more geological expertise to the author team. Showing multiple transects would be certainly needed for in-depth discussion of geology, but as we release the code with the published paper is hopefully something for future projects. The complexity of subglacial geology would ideally be presented with an application for evaluation in ice-sheet modelling to identify which parameters are most critical to predict the future evolution of the Polar ice-sheets. We hope that our manuscript is a contribution towards this direction by outlining the possibilities in using SOM and providing a first-order comparison with previously existing interpretations.

We will below address all the points raised by you in more detail.

**More**

**TITLE**- I find the current title a bit misleading and would therefore recommend changing it (see below)
Application of Self-Organizing Maps to characterize subglacial  bedrock properties in East Antarctica based on gravity, magnetic and  radar data
the paper does not give an overview or a critical discussion of bedrock properties for whole of East Antarctica; instead it essentially presents a new self-organising map derived from multiple existing datasets and their derivatives that potentially (although unfortunately not really discussed in the current MS) may provide a basis for new interpretations of subglacial bed geology, compared to the relatively simple classification of the study region in sedimentary basins (Type I and 2), crystalline basement and mixed bed previously proposed by Aitken et al. (2023) in their continent scale Antarctic analysis.

**Suggested revised title**

**Self-organizing map of the Wilkes and Aurora Subglacial Basin region in East Antarctica based on radar, aeromagnetic and aerogravity attributes**

We changed the title to  "Application of Self-Organizing Maps to characterize subglacial bedrock properties based on gravity, magnetic and radar data – An example for the Wilkes and Aurora Subglacial Basin region, East Antarctica", to indicate that we give emphasis on introducing the approach.

**Abstract-**

"Therefore, it is crucial to efficiently combine various attributes derived from satellite and airborne geophysical surveys to characterize subglacial properties"

**I find this claim difficult to follow**- what subgacial properties can we infer from satellites presently?- I would argue not many! We don't have high resolution views neither of bed topography nor of magnetic or gravity patterns from satellite data to image the heterogeneity of bed properties - but we do have a good large-scale view of the crust and lithosphere especially from GOCE (e.g. Ebbing et al., 2018; Pappa et al., 2019a,b). As this study does not deal with satellite data, I would suggest either dropping this sentence in the abstract -or state that while satellite data provides views of spatial and temporal changes in ice sheet dynamics, as well as glimpses into the deeper crust and lithosphere, airborne geophysical data are a key to resolve the heterogeneity in subglacial geology and its properties.

We agree and as we not use satellite data, we deleted the reference to those type of data.

"we evaluate a Self-Organizing Map (SOM), an unsupervised machine learning technique. The concept of SOMs, an unsupervised machine learning approach". **There is repetition here- please mention just once**
Adjusted.

"for the case at hand" **This could make a potential reader wonder other than a methodological application what the broader purpose of the study itself is.**
Changed, also in line with the other reviewer

Instead, I would strongly recommend highlighting earlier on in the abstract (right at the beginning) the key importance and impact of the study area- aka the Wilkes and Aurora Subglacial basins are huge marine-based sectors of the EAIS that could potentially be prone to dynamic instability in the future. Hence understanding the influence of the heterogeneity in subglacial geology and bed topography is particularly important here. Then I would say that the interpretation of these critical regions has so far relied on expert interpretations of airborne geophysical data but here you explore the application of a less subjective self-organising map as a tool to xxx etc.

As we concentrate on the introducing the approach, we did not add further details on the study area and moved this in the text, so that the reader has not a wrong impression and expects a full geological discussion. We did however, explain in more details the differences to earlier approaches.

"where also suitable data sets for the application of the SOM exist"
I would drop this sentence- we actually have many other areas in both East and West Antarctica where we have considerably better datasets than the ICECAP data used herein, which is mostly composed of very widely spaced radial lines (which is certainly NOT ideal for resolving the fine scale heterogeneity in subglacial geology and bed properties from aeromagnetic data for example- although it does give a large scale perpective of these huge marine based sectors of the EAIS that could not be achieved without a huge logistical effort if the surveys had been flown on conventional regularly spaced surveys such as those flown over the northern Wilkes Subglacial Basin (see also Ferraccioli et al., 2007, Terra Antartica Reports);

We changed accordingly.

Incidentally it is NOT just NASA data. The data you are using are from a major joint multinational US-UK-Australian and French and Italian effort ICECAP that was also BUT not exclusively supported by NASA! The data may have been downloaded from there but full recognition to ICECAP should also be given in the text.
"Previous analysis indicated the presence of both crystalline basement and sedimentary basins in the area, and our SOM shows a remarkable agreement"
**I would recommend rewording this**- highlight first what your approach brings to the table compared to the current state of the art- e.g. it unveils additional heterogeneity in the characterisation of cryptic properties of the subglacial highlands compared to continental-scale maps and then state that it aids depicting the distribution of sedimentary basins and crystalline basement domains in general agreement with previous expert interpretations based on aerogeophysical datasets;

As we have reworded the abstract, we think that providing first an agreement before we discuss differences is justified.

**incidentally what is new from this SOM map is actually pretty difficult to follow also because the lack of a sufficiently well-founded geological or geomorphological interpretation and discussion sections in the current MS.**
"These variations can potentially be exploited further in describing subglacial properties and the coupling between bed and overlying ice-sheets"-
**this raises the question of why this was not done here**.
I would drop this sentence in the abstract and perhaps leave this pointer pending for the discussion or conclusion.
**Unless the paper is re-written very extensively, I would suggest you focus here primarily on the new method and the map you deliver as a new tool mainly.**

As suggested we focus now on the method, less on the geological interpretation. The latter would change the focus of the paper, which is to provide a novel tool for interpretation of the Antarctic bed. However, we provide a new Figure 8 as zoom in so that it is easier to see the differences to the earlier study by Aitken et al. (2023).

**Introduction**

Aitken et al. 2023, Bell et al. 2008, McCormack et al. 2022. I would suggest adding Bingham et al., 2012, *Nature* & Jordan et al., 2023, *Science Advances* to the list here
We added these references

"airborne magnetic or gravity datasets". **I would prefer this to be reworded** into aeromagnetic and airborne gravity datasets.
Ferraccioli et al 2002, 2009
**I would recommend adding also** Ferraccioli et al., 2011, *Nature*;
**I would recommend adding also** Forsberg et al., 2018, doi: https://doi.org/10.1144/SP461.17
Reworded and added

That presents data from another critical marine-based sector of the EAIS;
McLean et al. 2009, Aitken et al. 2014, Kim et al. 2022
- Kim's paper il more about a compilation of magnetic data using satellite data- I would drop here.
Added and adjusted as advised.

"the interpretation requires some form of constraint to overcome the inherent ambiguities"
- this is of course true wrt to aeromagnetic data interpretation -but I would argue this also holds true if one uses multiple methods and their derivatives too.
For example, basal roughness varies at different scales and is not only due to subglacial geology, although at large spatial scales there will obviously tend to me a rougher bed over a crystalline-basement dominated Precambrian craton that a major sedimentary basin.
For the devil in the detail- see for example Jordan et al., 2010 Terra Nova. They show a nice example of an area with apparently quite uniform subglacial geology (at least according to potential field data interpretations) but remarkable differences in basal roughness (speculatively linked to a huge paleo-subglacial lake and associated drainage system);
Aeromagnetic and airborne gravity data may not resolve thin sedimentary drapes in marine basins (that is basins where marine incursion occurred in response to major ice sheet retreat atop of the pre-existing major sedimentary basins) at all;
Conversely, these data are more likely to image thick sedimentary basins that generally predate the EAIS by hundreds of millions of years- e.g. the Devonian to Jurassic Beacon Superbasin in the Wilkes Subglacial Basin region (see Ferraccioli et al., 2009a).
Where we have exposures of the rocks we can directly see that the mesa topography associated with Beacon Supergroup rocks massively intruded by Jurassic Ferrar sills tends to divert ice flow around it. Hence the patterns in the roughness of the bed are significantly more complex than a simple "smooth bed" atop of a sedimentary basin.
Overall, when one combines different methods it is important to address also how the individual limitations then combine together, also as a function of the features that one is then aiming to interpret in the first place (e.g. sedimentary basins, crystalline rocks-dominated mountain ranges etc).
And of course, one has to be careful-  there are huge mountain ranges dominated by sediments too that could have been incised by rivers and early glacial systems creating rough topography in highlands- see for example Rose et al., 2013 EPSL. A clear example of this are the Vostok Subglacial Highlands.
We rephrased this part and also referred to petrophysical data, as we wanted to express that constraints are necessary to tie the interpretation to the structures at depth. However. We are referring to the general ambiguity of potential field data, that require constraints. The cited paper by Betts et al. 2024 and Jordan et al. 2023 provide detailed explanation of the possibilities of using aeromagnetic data alone or in combination, respectively.

 "data sets" change to datasets
We have changed this.

"it is in part subject to interpretations"- well I would say that it is an interpretation!

We changed accordingly.

It should be clearer in the text that Alan's map is an inferred subglacial geology map showing the proposed distribution of sedimentary basins (Type 1 and 2 as Alan called them… a terminology that notably I still find a bit cryptic as a geologist), mixed bed and crystalline rocks-dominated bedrock
We have rewritten the paragraph.

"Please see Aitken et al. (2023) for more details on this".
I would not recommend using this type of sentence. Your paper needs to stand alone- and hence the main concepts and findings should be reviewed here- also because it is important to state more clearly what your own research objective is in the Wilkes and Aurora Subglacial Basin region. **Are you going to try and test and augment some of these previous interpretations with the aid a new tool?**
We have changed this.

"Machine learning and statistical based methods are nowadays popular approaches for less heterogenous models". **This sentence is rather unclear to me**. What are you trying to say exactly here? That we don't use machine learning for inferring subglacial geology or something else?
We rephrased this sentence.

**Self-Organizing Maps**- a bit more references here than a single reference from 1990! would have been helpful. Please provide more updated references.
We added two more references for geophysical applications. However, SOMs are not widely used as they have been so far computationally too demanding.

"The study area is in Wilkes Land, East Antarctica (Figure 1), chosen for the excellent coverage with line data"
**Firstly**, this is NOT only Wilkes Land. The area you covering includes Victoria Land, George V Land and Wilkes Land
**Secondly**, there is NOT excellent aeromagnetic or aeromagnetic data coverage here.
We agree and corrected. In line with the second reviewers' comments, we have rephrased this to explain that the area was convenient as the data sets were available. The reason for choosing the data was our expectation that the quality and preprocessing of the data would make the application straightforward. Coincidentally, the data evaluation was part of a data analysis for RINGS at NPI.

Instead, there are widely spaced reconnaissance ICECAP lines in the study region compared say to the many GITARA aeromagnetic surveys over Vitoria Land (see e.g. Ferraccioli & Bozzo, 2003- https://doi.org/10.1144/GSL.SP.2003.210.01.07 or Ferraccioli et al., 2009b- https://doi.org/10.1016/j.tecto.2008.11.028) or even compared to the WISE-ISODYN survey over the northern Wilkes Subglacial Basin (Ferraccioli et al., 2009; Jordan et al., 2013; https://doi.org/10.1016/j.tecto.2012.06.041), despite the key importance of this study region to comprehend the potential influence of heterogeneous bed topography properties on the behaviour of the marine based and hence potentially more vulnerable Wilkes and Aurora subglacial basin sectors of the EAIS.
Yes, we agree and hope that the approach will be used by others for other regions in East Antarctica.

What is important to state (and needs to be added in the text) is that despite the large coverage in terms of area the line spacing of ICECAP surveys is wide especially in the interior of the EAIS (where by the way the onset of ice streaming occurs);
We refrain from such a detailed discussion of the survey and line spacing as this would move the focus of the current paper. However, as a follow-up study, it would be interesting to explore how the results are changing depending on the line distance and spacing.

but your **SOM approach that exploits the full potential of line data still adds new views of the region when compared to interpretation methods based on grids alone**
We rephrased the part to provide more explanation

Interested reader- I think a little bit more background would have helped here;
We added more details.

3.1 Study area
I find this is a rather incomplete description of the study area.
The various bedrock features, basins, and geological domains should come across much more clearly here, especially if one then wants to highlight later on that the SOM approach provides hints into further complexities than previously inferred.
And as noted above -the coverage is reconnaissance in character with very widely spaced lines especially in the interior of the EAIS due to the radial flight pattern of ICECAP
"consequently its massive potential for sea-level rise" – You need some references here!
(Fretwell et al., 2013)- is clearly NOT a good reference for the hot debate on the stability of this part of the EAIS. Please do the relevant literature research and cite the relevant papers.
And one needs to be updated to also add BEDMAP 3 that incorporates all the new data in this region!
(references Pritchard et al., 2025, https://doi.org/10.1038/s41597-025-04672-y; Fremand et al., 2023, https://doi.org/10.5194/essd-15-2695-2023) including in the figure.
In line with the suggestions by the other reviewer, we removed this part.

**3.2. Datasets**
We use the NASA Operation Ice Bridge (OIB) dataset collected between 2009 and 2012- ICECAP needs citing too!
We cited the data source.

Magnetic anomaly (taken from ADMAP-2 (Golynsky et al. 2018)?
I am bit puzzled here- you must have used a subset of ADMAP 2 here as we have a lot more aeromagnetic data in the study region- I presume you only used the re-processed ICECAP radial profiles we then compiled in ADMAP 2.
We explain in the text, that we use the database, not the grid and only the lines where we have radar data.

I suggest you use a transparent backdrop for all the datasets with BEDMAP 3, ADMAP 2 and the AntGG compilation and show the ICECAP radial lines you are re-analysing atop of the backdrop.
We changed Figure 1

Effectively, despite all these additional data that would clearly help contextualise the geological and bed interpretation processes along the profiles you are going back solely to the individual ICECAP profiles which I find makes the paper mostly methodological focused
Yes, and we kept that focus based on the recommendations by the other reviewer.

*Figure 2.* The reversed magnetic anomaly scale with blue tones over highs and browns over lows is rather unconventional. I find it puzzling when compared to the original ADMAP 2 data or the input data of Aitken et al., 2014. Please change this!
We have adopted the color scale also following the advice of the other reviewer to use color-blind friendly color schemes.

Also, I wonder why the authors are not using the Ebbing et al., 2021 product that also replaces the long-wavelength component of the magnetic field with SWARM satellite magnetic data. By the way ICECAP data did not match very well higher resolution and better levelled data on the TAM side.
We extracted the magnetic data from the ADMAP 2 line data base. In this sense, we did not have to resample the data long the flight lines, which would have been necessary, if we used Ebbing et al.2021. Furthermore, the SOM is sensitive to near surface structure sin the current set-up, not

sensitive to long-wavelength magnetic field. Furthermore, as seen inEbbing et al. 2021, the correction by replacing the long-wavelength part of ADMAP-2 grid with satellite data is small.

For the gravity data- I find the inverted colour scale with gravity lows shown in red confusing. Please change this!
We changed the color scale.

**Airborne gravity data;** MacGregor et al. 2021 etc
If you asked NASA and the ICECAP team they would likely have given you the data.
We did enquire about the data and could not get access to an improved, processed data set

Also one has to be careful here. We did not have as good bedrock topography data compared to BEDMAP 3 at the time that we made the AntGG compilation in 2014 and then published it in 2016 in the Scheinert et al., 2016 GRL paper so ideally one would have:
1. Accessed the complete Free-Air gravity data
2. Recomputed the Bouguer anomaly based on the updated BEDMAP 3 compilation
Anyhow, if this is not possible at this stage to do this extra work, at least some sentence on the availability of more updated bed topography and ice thickness datasets in particular for Bouguer anomaly calculation should be made
We discuss in the text, that it would be preferential to use line data, nut use a gridded product.
"Instead of using the bedrock topography, it can be useful to determine the isostatic adjusted topography tiso".
I disagree it is not instead- BOTH the original bed and a rebounded version of the bed should be used and discussed.
Yes, both data sets could have been used and of course, there is geological information contained. The reason for using the isostatically corrected topography was that we are mainly interested in short-wavelength features of topography, less by regional trends. Essentially, we only removed a long-wavelength trend, that we considered not relevant here.

That said the bed is NOT going to rebound in an Airy way when the EAIS is removed (that is essentially prior to its formation -as the EAIS is unlikely to have ever completely collapsed since its formation) or retreated –
because these are typically more complex visco-plastic process than predicted by an Airy model.
Some explanation on the simplifications associated with the Airy model should be introduced. And one should explain for the non-expert reader that the Bouguer anomaly map is strongly influenced by crustal thickness variations and hence the use of Airy isostatic maps can help enhance the shallow level intra-crustal sources of the anomalies.
In addition, if one wants to make more meaningful comparisons with **pre-glacial topography and geology then one should consider that huge amounts of glacial erosion and sediment unloading**and flexure have occurred particularly in the region of these subglacial basins since EAIS initiation at the Eocene-Oligocene boundary – **so an interesting product to use would have been to also use the** Paxman et al 2019, 2020 **paleotopography datasets** (https://doi.org/10.1016/j.palaeo.2019.109346; https://doi.org/10.1029/2020GL090003) **as this provides a more realistic view of the pre-glacial landscape and its links with pre-glacial geological boundary conditions**
We agree, but as we stated, we limit ourselves to a selection of attributes. Off course, different types of data are available, but we limit us to the data sets conventionally used for interpreting subglacial properties. The codes will be made available and we invite the community to further exploit the use of SOMs.

**Attributes**
When you refer to the attributes please refer to the table upfront.
Done
I would much have preferred to see some more profile views (e.g. at least a couple per basin and flanking mountain ranges) of the major features with multiple individual attributes displayed first and

tied much better to a map view with a clearer overlay of the major subglacial basins and highlands and geological domains that one is investigating.
This is essential for any interpretation and discussion of the features.
*Again, we apologize that the reviewer might have been misled by us. Our contribution is introducing the possibilities of using SOMs in ice-covered regions, not a full geological interpretation.*

*Figure 3 Example of attributes used for the SOM.* The logic of this example is unclear to me. Why these rather than many others?
*We added in the revised version figures of all attributes as supplementary material. The main reason for not showing all here is, that the details are only of interest for experts, not the general reader.*

**B Spectral Power-** is this not influenced by distance to source?- the pink colours seem to correspond to outcropping Precambrian crystalline basement rocks
See explanation in text
**3C** I find again the inverted colour scale un-conventional and the meaning of the shape index is also unclear to me
We changed the color scale. The meaning of the shape index is explained in Lie et al. (2015) and Ebbing et al. (2018) as indicated in the text

I believe it would have been better to present maps for related quantities fist, aka bed, magnetic and gravity followed by mixed attributes for potential correlations (or lack thereof).
We show all input data in Figure 3, examples of attributes in Figure 4 and all attributes as supplementary material. Mixed attributes is not something we can provide as the idea of the self-organizing map is that this is user intervention independent. Off course, there exist other approaches, where mixed attributes are calculated (e.g. Henkel petrophysical plot based on inversion results: e.g., Enkin et al. 2020 https://doi.org/10.1029/2019GC008818), but these approaches are based on a user choice of input data. Showing all possible mixed attributes is not possible due to the number of possibilities, hence the choice of SOM.

Some explanation of why the spectral centroid bed is used would be useful- what can this yield?
We added a sentence.
ξ is the integrated power spectral density of the bed elevation profile in a 500 m to 2000 m wavelength bin.in a 500 m to 2000 m wavelength bin.
Why was this wavelength used? Is there a wavelength dependency and what can different wavelengths be representative of?
We use a bin covering 700 m to 800 m lag distance.  Why and what impact does this choice have?
we used a 10 km-moving average filtered bed elevation. Why was 10 km chosen?
The TDX signal is the tilt derivative of the magnetic field. Ok but what effect does the tilt derivative have on the aeromagnetic data- and what was the purpose of this enhancement?
I assume the aim was enhancing the contribution of shallower magnetic anomaly sources as a way of helping in defining the heterogeneity in the bedrock geology? Needs explaining for the non-expert reader.
The choice was made according to typical parameter choices to enhance sensitivity to near-surface features as we explain in the text. Again, Betts et al. (2025) discuss the possibilities of using such filters, but if unjustified, the SOM will automatically ignore such attributes by giving them a lesser weight.

*Moving average filtered magnetic anomaly (Mean mag)*
Enhancing the short wavelength content- explain why this is useful
*Analytical signal (AS mag)*
When using higher order derivatives a careful analysis of the noise should be provided. Even where much higher quality aeromagnetic data on regular grids are available e.g over northern Victoria Land (e.g. Ferraccioli et al., 2009b) we generally had to filter the data with Hanning or other filters so as not to have too much noise in AS products.
We discuss in the text, that some features might relate to data noise.

*Isostatic anomaly (Iso grv)*

I would have preferred the use of BEDMAP 3 that is based on line data as opposed to a product that includes ice sheet mass conservation such as BedMachine. Additionally, the density contrast used seems very high for East Antarctica 530 kg/m3.

As we explain in the text, different data sets could be explored, but we mainly use the radar data to derive the attributes.

*Analytical signal (AS grv)*

Again- with the higher order derivatives it is important to check that the noise levels are not rendering noise in the grid only. Also, it is important to note that the definition of out of plane bedrock topography and ice thickness outside the widely spaced survey lines is poor and hence in the Bouguer anomaly data itself there can already be quite a lot of noise that can tend to blow up with the high order derivatives. Some smoothing/filtering is recommended

The regional grid by Scheinert et al. (2016) does not show a high a noise level and we made sure not to introduce artefacts with the equivalent source method applied. However, that might become an issue, when using line gravity data.

**3.4 SOM calculation, used algorithms and other considerations**

Please provide a simpler and less vague (..other considerations) heading

e.g. SOM calculations is fine where limitations etc can be addressed in the text perhaps as a sub paragraph. The resulting SOM has a shape of 30 by 30- what?

Changed accordingly

**4 Results and discussion**

I would have preferred to have seen a more critical and focussed discussion presented and preceded by an interpretation

We do focus now even more on the methodological development in line with the comments by the other reviewer.

"Such a varying topography will cause a varying gravity and to a minor portion magnetic signal".

**I am not sure I understand this statement.**

Apologies. We corrected the sentence

High frequency magnetic anomalies will correlate with rough crystalline basement topography. I don't understand why this would be considered surprising. Say you go over northern Victoria Land where you see the rough bed associated with relatively well exposed granitoids and metamorphic rocks and you clearly appreciate in tilt derivative maps the high frequency anomalies that are caused by these rocks. There are of course some exceptions where the rock assemblages are magnetite-poor but at large-scale this holds.

Yes, that is not an observation, but meant as an explanation. We rephrased.

"That corresponds to the different sensitivity of the gravity and magnetic field to the sources, but might also indicate that we miss some of the gravity signal by using a gridded data set as input and not measurements along the flight lines"

That is not likely to be the only reason. It is important to note that airborne gravity data are heavily filtered typically with a ca 9 km half width filters (see e.g. Jordan et al., 2013- https://doi.org/10.1016/j.tecto.2012.06.041)

We agree, but still the gridded gravity data product has a relatively low resolution.

4.2 Weights for individual attributes

Before we analyse the SOM in more details with respect to its possible (geological meaning) please take the parenthesis out of the sentence does not make sense

*Figure 5 Weights for every attribute and cell of the SOM. All attributes where rescaled using the standard deviation, before the training started.*

I am not sure I understand this figure.

We rewrote this paragraph and added a sentence to the Figure Caption. Hopefully now it is clearer that

this is not a geographical representation

**Figure 6 Visualization of the SOM and class distribution**
I find this figure quite difficult to follow.
Before having everything displayed together it may perhaps have been clearer to see each category of features proposed by Aitken et al., 2023 displayed individually (and ideally also with selected examples extracted in the correct geographic location) vs the SOM.
That is following in Figure 7 and the new figure 8.

Aka to what extent can one define for different areas Type 1 basins, Type 2 basins, mixed beds etc using a SOM approach? (also considering large differences between the widely spaced interior regions and the coastal sectors where coverage is much better).
The approach is independent of the line distanced as we use only data long the flight lines for our analysis.

I am also struggling a lot to have a spatial view. I assume that although we are seeing a map this is not locating the corresponding features in map view.
This is not a geographical map, but the SOM itself. The dimension is the elements and we sampled for each domain the agreement with the map by Aitken et al.

I also don't understand the letters and the dashed lines that well. For example, when I look at region E there seems to be a remarkable contrast between the red and blue regions with little similarity between them -so I fail to understand while this all labelled as a single feature E.
This is a subjective choice and could certainly be improved by applying an automated classification scheme. We added some more explanation in the text

*Figure 7: Representation of the clusters from the SOM. A) Geographical distribution of the SOM., B) Unified distance matrix for the presented SOM. C) Classification of geological bed type from Aitken et al. (2023). Yellow line indicates profile of Figure 8.*
This figure is difficult to follow. Firstly, the colour scale is very difficult to follow as the differences between the colours between A and C and D and E is hard to appreciate. I am not sure the text has explained clearly enough what the unified distance matrix shown in panel b is and panel c has no km scale or coordinate label. It is critical that you show the previous interpretation of Aitken et al., 2023 with labels of the geographic features you then refer to in the text at the SAME scale as the colour map of the SOM. Furthermore, I would suggest that you show an additional panel with these overlain so one can start thinking about the potential interpretation of additional heterogeneities not addressed in the continental scale interpretation of Aitken et al. Also where is the yellow line showing the location of Figure 8?
The colormap for A) is 2 dimensional. We make this choice to be able to use a SOM which is reducing to 2 Dimensions instead of only to one Dimension. This is rather complex and there is no out of the box solution we could use. We here vary the red and blue channel at x and y dimension respectively, while keeping the green channel at 0. This choice is specifically designed to work for people with red–green color blindness. While there would be other possible combinations in the RGB space, we assesed this one to provide the best contrast. For B) we indicated that this is the unified distant matrix for the 30x30 cells of the SOM. We also added a new Figure 8 to allow more details in the comparison.

**Figure 8 should also report locations. What basin and highlands are we looking at?**
We added geographical locations.
I would have preferred at least a couple of profiles per major sublglacial basin and a more detailed discussion of results -not just the methodological part
This is understandable, but would change significantly the focus of the manuscript. That is not on the geological interpretation, but the availability of a new tool, demonstrated with a few examples. A full geological analysis would be interesting, but as stated by you, would require testing of alternative data sets and so on, which is well beyond the scope of the paper.

---

## Author Comment (AC2)

Reply to reviewer Tobias Stål

We thank the reviewer for the insightful comments and the annotated manuscript. We followed the advice and concentrated on presenting the approach of SOMs, giving less emphasis on the geological interpretation. Hence, major changes in the manuscript are in the Introduction, Discussion and Conclusions chapter as advised by the reviewer and added further details by reorganizing the figures, adding a new zoom in for comparison and supplementary figures showing all attributes used for the analysis.

The paper demonstrates several notable strengths, particularly in its methodological approach. The use of self-organizing maps (SOM) stands out as a positive aspect. SOMs are well-suited to combine multivariate data into meaningful clusters. Compared to more traditional methods, SOMs often excel at revealing underlying patterns and relationships. The authors use a reasonable range of attributes to identify such patterns, thereby capturing the nature of the subglacial geology in Wilkes Land, East Antarctica. Furthermore, SOMs operate in an unsupervised manner, reducing potential bias from manual classification and handling noisy or incomplete data more robustly than some alternative approaches. Their transparent structure and intuitive results make them especially valuable for insights into the structure of heterogeneous data, where the interpretability and simplicity of SOMs can be more beneficial than the raw predictive power of deeper, more complex models. The subglacial Antarctica serves as an experimental sandbox for method development, and we certainly urgently need to get a better understanding of the tectonic structure to provide ice sheet models with boundary conditions that define the ice sheet's stability over the coming centuries.

Thank you for your interest in the method. In line with your comments below, we restructured the manuscript to give emphasis on introducing the method of SOM for subglacial imaging.

Another strength of the paper is the decision to (mainly) work directly with flight line data rather than relying on interpolated grids. This approach preserves the original measurement fidelity and avoids the artifacts or smoothing effects. By analysing the flight line topography data directly, the study ensures that its results more accurately reflect the true spatial variability, which is crucial when dealing with complex or heterogeneous terrain. The discussion regarding the attributes and their relationship to geology is insightful and engaging.

Thank you. And we followed your advice to concentrate on explain the approach and not to extend the geological interpretation.

Despite these methodological strengths, the paper currently suffers from weaknesses in its writing and presentation. The manuscript often lacks clarity, making it challenging for readers to follow the narrative. The overall structure feels incomplete, and the paper appears to have undergone insufficient editing prior to submission—it reads more like a first draft than a finished manuscript. As a result, key points and the significance of the findings are sometimes difficult to discern. In particular, the Introduction, Abstract, and Discussion would benefit from substantial revision, while the more mathematically driven Methods and Results sections are in better shape. The figures and tables are generally okay and useful; however, I strongly recommend changing the colour maps, as they are not perceptually uniform.

The color schemes were changed and are hopefully better suited. Regarding the comments on figure 7 in the annotated manuscript, the Colormap is 2 dimensional. We make this choice to be

able to use a SOM which is reducing to 2 Dimensions instead of only to one Dimension. This is rather complex and there is no out of the box solution we could use. We here vary the red and blue channel at x and y dimension respectively, while keeping the green channel at 0. This choice is specifically designed to work for people with red–green color blindness. While there would be other possible combinations in the RGB space, we chose the current one as it appears to provide the best contrast.

To strengthen the paper, I recommend a thorough revision to improve clarity, ensuring that the background, objectives, and discussion are well explained and logically organized, and that any unnecessary digressions are avoided. Careful attention to language and grammar will also help bring the manuscript up to the journal's standards. Additionally, the paper would benefit from a bit more context and justification for the choice of methods, particularly in clarifying how Self-Organizing SOMs) differ from other potential approaches. That said, I agree that an extensive theoretical background is not required here.

We added more explanation on the choice of method and the differences to alternative methods in the introduction and carefully checked the English (US) grammar.

In summary, while the methodological choices are strong, the manuscript requires significant revision for clarity and completeness before it can be considered suitable for publication. I am providing an edited and commented Word file (with apologies for any formatting issues caused by converting from PDF). My suggestions are not exhaustive, but I hope they offer a useful starting point for revising the manuscript and addressing some of the recurring issues. Additionally, some edits may have altered the authors' intentions and should be disregarded, but they are helpful in highlighting where the text is unclear.

Thank you for your comments in the annotated manuscript, which we followed as much as possible. We especially rewrote the Discussion and Conclusions chapters to have a clearer narrative, concentrating on introducing the SOM approach.

In the manuscript with track changes, it will be clear how we implemented these changes to restructure and to rewrite the manuscript, making it hopefully suitable for publication.

---

## Author Response (AR2)

We thank the reviewer and editor for the comments and suggestions for improvement.

We have once more carefully checked the manuscript and hopefully improved the readability.

We have also asked a native speaker to flag any unnecessarily convoluted sentences, which resulted in minor changes and grammar corrections all over the manuscript and have added the link to the juypter notebook and training data sets on Zenodo.

We hope that the current version is now acceptable for publication.

On behalf of the authors

Jörg Ebbing